# The Deubiquitinating Enzyme AMSH1 Contributes to Plant Immunity Through Regulating the Stability of BDA1

**DOI:** 10.3390/plants14030429

**Published:** 2025-02-01

**Authors:** Yiran Wang, Weijie Huang, Xin Li, Yuelin Zhang

**Affiliations:** 1Department of Botany, University of British Columbia, Vancouver, BC V6T 1Z4, Canada; yiran.wang@botany.ubc.ca (Y.W.); weijie.huang@msl.ubc.ca (W.H.); 2Michael Smith Laboratories, University of British Columbia, Vancouver, BC V6T 1Z4, Canada

**Keywords:** plant immunity, SNC2, BDA1, deubiquitinating enzyme, AMSH1, *Arabidopsis*

## Abstract

Plants utilize plasma membrane localized receptors like kinases (RLKs) or receptor-like proteins (RLPs) to recognize pathogens and activate pattern-triggered immunity (PTI) responses. A gain-of-function mutation in the *Arabidopsis* RLP *SNC2 (SUPPRESSOR OF NPR1-1, CONSTITUTIVE 2)* leads to constitutive activation of defense responses in *snc2-1D* mutant plants. Transcription factors, SYSTEMIC ACQUIRED RESISTANCE DEFICIENT 1 (SARD1) and CALMODULIN-BINDING PROTEIN 60g (CBP60g), define two parallel pathways downstream of SNC2. The autoimmunity of *snc2-1D* was partially affected by single mutations in *SARD1* or *CBP60g* but completely suppressed by the *sard1 cbp60g* double mutant. From a suppressor screen using *sard1-1 snc2-1D*, we identified a deubiquitinating enzyme ASSOCIATED MOLECULE WITH THE SH3 DOMAIN OF STAM 1 (AMSH1) as a key component in SNC2-mediated plant immunity. A loss-of-function mutation in *AMSH1* can suppress the autoimmune responses of *sard1-1 snc2-1D*. In eukaryotes, selective protein degradation often occurs through the ubiquitination/deubiquitination system. The deubiquitinating enzymes that remove ubiquitin from target proteins play essential roles in controlling the level of target protein ubiquitination and degradation. As loss of *AMSH1* results in decreased BDA1 abundance and BDA1 is a transmembrane protein required for SNC2-mediated immunity, AMSH1 likely contributes to immunity regulation through controlling BDA1 stability.

## 1. Introduction

Plants have evolved effective defense mechanisms to thwart the surrounding microbial pathogens. Upon pathogen detection, the initial defense of plants begins with the recognition of evolutionarily conserved pathogen-associated molecular patterns (PAMPs) by pattern recognition receptors (PRRs) at the cell surface, leading to pattern-triggered immunity (PTI) [1]. Successful pathogens are able to suppress PTI through secreted effectors, which results in effector-triggered susceptibility (ETS) [2]. In turn, higher plants use intracellular nucleotide-binding leucine-rich repeat receptors (NLRs) to recognize cognate effectors and turn on effector-triggered immunity (ETI) [2]. Activation of PTI or ETI results in a number of overlapping downstream outputs, such as activation of mitogen-activated protein kinase (MAPK) cascades, calcium flux, reactive oxygen species (ROS) burst, callose deposition, transcriptional reprograming, and phytohormone signaling [3].

PRRs are usually either receptor-like kinases (RLKs) or receptor-like proteins (RLPs) that lack an intercellular kinase domain [4]. *Arabidopsis SNC2 (SUPPRESSOR OF NPR1-1, CONSTITUTIVE 2)* encodes an RLP. Loss of *SNC2* results in enhanced susceptibility to pathogenic bacteria *Pseudomonas syringae* pv *tomato* (*Pst*) DC3000, demonstrating that SNC2 is essential for basal resistance [5]. The alteration of a Gly to Arg in a conserved motif of the transmembrane domain in *snc2-1D* results in smaller size and constitutively activated defense responses including elevated defense marker gene expression as well as defense hormone salicylic acid (SA) accumulation. Studies on *snc2-1D*-mediated autoimmunity revealed that BDA1 (for Bian Da; “becoming big” in Chinese), a novel ankyrin-repeat transmembrane protein, functions downstream of SNC2 to regulate plant immunity [6]. Knocking out *BDA1* leads to full suppression of *snc2-1D* autoimmunity, whereas a gain-of-function allele of *BDA1*, *bda1-17D*, exhibits constitutively activated cell death and defense responses.

Two members of the *Arabidopsis CALMODULIN-BINDING PROTEIN60* (*CBP60*) gene family, *CBP60g* and *SYSTEMIC ACQUIRED RESISTANCE DEFICIENT 1* (*SARD1*), encode positive regulators of plant immunity that promote the pathogen-induced SA production [7,8]. Both genes are induced by virulent bacterial pathogen *Pseudomonas syringae pv maculicola* (*Psm*) ES4326 infection. Single mutants of either gene only exhibit modest increases in the growth of *Psm* ES4326. However, *sard1 cbp60g* double mutants allow greatly increased growth of *Psm* ES4326 and reduced levels of pathogen-induced SA [7,9]. Furthermore, Chromatin immunoprecipitation (ChIP)-seq analysis revealed that the expression of a battery of genes encoding key regulators of PTI and ETI is directly controlled by SARD1 and CBP60g during plant defense, demonstrating that SARD1 and CBP60g function as master regulators of plant immunity [10].

A previous report showed that SARD1 and CBP60g contribute to autoimmunity of *snc2-1D* [10]. The dwarf stature as well as constitutively activated immune responses are partially suppressed in *sard1-1 snc2-1D* and *cbp60g-1 snc2-1D*, and they are almost fully blocked in *cbp60g-1 sard1-1 snc2-1D*. These findings indicate that SARD1 and CBP60g define two parallel pathways downstream of SNC2 to mediate defense responses [10]. However, how SNC2 activates SARD1 and CBP60g requires further investigation.

Plants use various ways to maintain protein homeostasis. One mechanism is through ubiquitin (Ub)-mediated protein degradation. Ubiquitination is a reversible process, where the bond between Ub and the substrate can be cleaved by deubiquitinating enzymes (DUBs) to remove and rewrite the Ub modifications [11]. There are around 50 DUBs encoded in the *Arabidopsis* genome, which can be classified into seven protein families according to the differences in their catalytic domains [12]. Different from the other six subfamilies belonging to cysteine proteases, the JAB1/MPN/MOV34 (JAMM) subfamily uses a zinc metalloproteinase domain for cleavage [13]. RPN11, a DUB from the JAMM subfamily with an MPN+ domain (MPR1, PAD1 N-terminal+), has been implicated in hydrolyzing Ub chains prior to protein degradation by the 26S proteasome [14]. Additional MPN+ domain proteins, AMSH1 and AMSH3, were shown to be necessary for autophagic and endocytic degradation, respectively [15,16,17,18]. However, due to their broad target spectra, the specific substrates and physiological role of most DUBs are poorly defined.

To identify regulators required for CBP60g-dependent defense responses downstream of SNC2, a suppressor screen was performed in the *sard1-1 snc2-1D* background. Mutations in components functioning downstream of SNC2 in the CBP60g-dependent signaling pathway would result in the suppression of *sard1-1 snc2-1D*’s autoimmunity. Here, we report the identification and characterization of a suppressor 123-1 mutant. 123-1 carries a mutation in *AT1G48790*, which encodes a DUB ASSOCIATED MOLECULE WITH THE SH3 DOMAIN OF STAM 1 (AMSH1). The DUB activity of AMSH1 toward Lys 63-linked Ub chains has been previously confirmed in vitro [17]. Our current analysis showed that AMSH1 is involved in regulating SNC2-mediated immune responses by affecting the protein level of BDA1.

## 2. Results

### 2.1. Identification and Characterization of Suppressor Mutant 123-1

A forward genetic screen was conducted in *sard1-1 snc2-1D* to identify regulators required for CBP60g-dependent defense responses downstream of SNC2. Mutants with suppressed dwarf morphology of *sard1-1 snc2-1D* were isolated. 123-1 was one of the mutants recovered. As shown in Figure 1A, the 123-1 mutants displayed intermediate plant size compared to wild-type Col-0 and *sard1-1 snc2-1D*. The *Pathogenesis-Related* (*PR*) gene *PR1* encodes a cysteine-rich secreted proteins with antimicrobial activities, while *PR2* encodes cell-wall degrading beta 1,3-glucanase. Both genes are usually highly induced during defense, and they have been used as defense marker genes. Quantitative reverse transcript-PCR (qRT-PCR) analysis showed that the constitutive expression of *PR1* and *PR2* in *sard1-1 snc2-1D* was largely suppressed in 123-1 (Figure 1B,C). In addition, SA level in the 123-1 mutants was greatly reduced compared with *sard1-1 snc2-1D* (Figure 1D). Moreover, the enhanced disease resistance against the oomycete pathogen *Hyaloperonospora arabidopsidis* (*Hpa*) Noco2 in *sard1-1 snc2-1D* was severely compromised in the 123-1 mutant (Figure 1E). Together, these data indicate that 123-1 suppresses the autoimmunity of *sard1-1 snc2-1D*.

### 2.2. Mapping-by-Sequencing of 123-1

To identify the causal mutation in 123-1, whole genome sequencing (WGS) was performed on a 123-1 mapping population. When 123-1 was backcrossed to *sard1-1 snc2-1D*, the resulting F1 plants displayed *sard1-1 snc2-1D*-like morphology (Appendix A). Furthermore, in the F2 progeny, a 48:136 segregation of suppressor-like and *sard1-1 snc2-1D*-like plants was observed, which matches to the expected 1:3 ratio (χ^2^ = 0.1127; *p* value = 0.7375), suggesting that the autoimmune suppressing phenotypes of 123-1 are due to a single recessive nuclear mutation. F3 seeds from the suppressor-like F2 plants were harvested and planted. Tissue was collected for DNA extraction from 34 confirmed homozygous F3 populations. Following WGS and data analysis, a genetic linkage region was identified on Chromosome 1 (Appendix A). In this region, one of the candidate genes, *At1g48790* (encoding AMSH1), contains a G to A mutation at the splice junction of the tenth exon.

### 2.3. Loss of AMSH1 Suppresses the Autoimmunity of sard1-1 snc2-1D

To determine whether the mutant phenotype of 123-1 is caused by the mutation in *AMSH1*, a CRISPR/Cas9 construct with two gRNAs targeting *AMSH1* was generated and transformed into the *sard1-1 snc2-1D* plants. Two independent CRISPR lines were recovered with deletions inside the gene (Figure 2A). Similarly to the original suppressor mutant 123-1, these deletion lines displayed a larger size than *sard1-1 snc2-1D* (Figure 2B). The constitutive expression of *PR1* and *PR2* was suppressed in these CRISPR lines as well (Figure 2C,D). Consistently, the elevated SA accumulation as well as enhanced disease resistance against *Hpa* Noco2 was blocked in these lines (Figure 2E,F).

Lastly, the wild-type *AMSH1* under its own promoter was transformed into the 123-1 mutant. As shown in Appendix A, *AMSH1* complemented the 123-1 phenotypes (Figure 2F). Taken together, the CRISPR and complementation data suggest that the *amsh1* mutation in 123-1 is the causal one for the suppressor, and *AMSH1* is required for the autoimmunity of *sard1-1 snc2-1D*.

### 2.4. amsh1 Single Mutants Exhibit Compromised PTI Responses and Basal Immunity

To dissect the role of AMSH1 in plant immunity, infection experiments were performed on one *amsh1* T-DNA insertion allele (referred to as *amsh1-1* in this study) and one CRISPR/Cas9 deletion allele (referred to as *amsh1-2* in this study) in the Col-0 background. Considering the requirement of AMSH1 in RLP SNC2-mediated immunity, wild type plants and *amsh1* mutants were challenged with the type III secretion system-deficient pathogen *Pst* DC3000 *hrcC*, which triggers PTI. As shown in Figure 3A,B, the differences in basal *PR1* and *SARD1* expression levels in wild types and *amsh1* mutants were barely detectable with mock treatment. However, upon *Pst* DC3000 *hrcC* infiltration, *amsh1* mutants displayed significantly lower expression levels of *PR1* and *SARD1* than wild-type plants. Consistently, *amsh1* mutants supported more *Pst* DC3000 *hrcC* growth than wild-type plants (Figure 3C), suggesting that *amsh1* mutants exhibit compromised PTI. In addition, when *Pst* DC3000 was used to inoculate wild types and *amsh1* mutants, the enhanced susceptibilities of *amsh1* mutants against this bacterial pathogen were more obvious (Figure 3D). Therefore, consistent with the attenuated PTI responses, *amsh1* mutants also exhibit compromised basal immunity.

### 2.5. AMSH1 Regulates BDA1 Protein Levels

AMSH1 is an active DUB. Its catalytic MPN+ domain specifically cleaves Lys 63-linked polyubiquitin chains from substrates to release both the Ub and the target, preventing the degradation of the pre-targeted protein [13]. Considering the requirement for AMSH1 in the constitutive defense responses of *sard1-1 snc2-1D*, we hypothesized that AMSH1 may prevent the degradation of key regulators in SNC2-mediated immunity, such as BDA1 [6]. To test this, we silenced the *AMSH1* homolog in *Nicotiana (N.) benthamiana* and examined whether the protein level of BDA1 was affected. As shown in Figure 4A, comparing with the empty vector control, co-infiltration of the RNAi construct carrying the *NbAMSH1* fragment with the construct expressing BDA1-HATurboID resulted in decreased BDA1 levels in *N. benthamiana* leaves, suggesting the possible role of AMSH1 in preventing BDA1 degradation. Furthermore, we tested whether loss of AMSH1 function leads to decreased accumulation of BDA1 by crossing *amsh1-1* into the BDA1-GFP transgenic line. Western blot analysis showed that BDA1-GFP protein levels dramatically decreased in *amsh1-1* (Figure 4B). Since the differences in BDA1 transcript levels in *amsh1-1* and transgenic lines was barely detectable (Figure 4C), the decrease in BDA1-GFP protein level is most likely due to decreased BDA1 stability in *amsh1-1*.

## 3. Discussion

From the suppressor screen of *sard1-1 snc2-1D* to identify regulators involved in CBP60g-dependent resistance pathways downstream of SNC2, we found that the DUB AMSH1 contributes to *snc2-1D* autoimmunity. Mutation in *AMSH1* leads to suppression of the autoimmune responses in *sard1-1 snc2-1D* plants, including elevated *PR* gene expression, SA accumulation, and enhanced disease resistance (Figure 1 and Figure 2). Accordingly, the *amsh1* single mutant exhibits compromised PTI responses and basal resistance against bacterial pathogens (Figure 3). The constitutive defense responses of *snc2-1D* specifically requires BDA1, a transmembrane protein with unknown function [6]. Loss of *AMSH1* in *N. benthamiana* or *Arabidopsis* results in less BDA1 protein levels (Figure 4), suggesting that AMSH1 may contribute to SNC2-mediated immunity by affecting BDA1 protein abundance (Figure 5).

The identified point mutation of *AMSH1* in 123-1 locates at the splice junction of its tenth exon. It likely affects the splicing of *AMSH1*, leading to the retention of the nineth intron, which would ultimately result in producing a truncated AMSH1 protein without the catalytic MPN+ domain (Appendix A). The MPN+ domain is highly conserved between AMSH1 and AMSH3. It is necessary for the cleavage of Lys 63-linked polyubiquitin chains from substrates, releasing both the Ub and the target protein [13]. Single amino acid substitution in the MPN+ domain of AMSH3 abolish the deubiquitinase activity of AMSH3 [19]. In the 123-1 mutant, the mutated AMSH1 protein probably fails to deubiquitinate its substrate, leading to reduced BDA1 levels.

Whether BDA1 is a direct substrate of the DUB AMSH1 requires further investigation. Previously, Katsiarimpa et al. proposed two possibilities for AMSH1’s function in plant immunity: either SA-dependent basal immunity and cell death are indirectly affected in the *Ler amsh1* mutant or AMSH1 directly regulates plant defense [17]. If BDA1 is a substrate of AMSH1, it would provide good support for the second hypothesis.

Although BDA1-GFP was barely detected in *amsh1-1* (Figure 4B), loss of *AMSH1* only partially suppressed *sard1-1 snc2-1D*’s autoimmunity (Figure 1 and Figure 2), while loss of *BDA1* leads to full suppression of *snc2-1D* [6]. This could represent the combined effects from different substrates of AMSH1. On the other hand, it may be due to the role of AMSH1 in regulating intracellular trafficking and degradation of endocytic cargoes [12]. Studies have shown that ubiquitination/deubiquitination processes contribute to the regulation of endocytosis and the subsequent vacuolar degradation of plasma membrane receptors in plants, including REQUIRES HIGH BORON 1 (BOR1), FLAGELLIN-SENSITIVE 2 (FLS2), and PIN-FORMED 2 (PIN2) [20,21,22]. *amsh1* mutants are defective in Endosomal Sorting Complexes Required for Transport (ESCRT)-mediated vacuolar targeting [17]. Specifically, enzymatically inactive AMSH1 inhibits AvrPtoB-dependent endocytic degradation of CHITIN ELICITOR RECEPTOR KINASE 1 (CERK1) [18]. It remains to be determined whether AMSH1 contributes to the endocytic degradation of SNC2 or even its paired unknown RLK or BDA1.

The *Arabidopsis* genome encodes three AMSH family members, namely AMSH1, AMSH2, and AMSH3. The amino acid sequences of AMSH1 and AMSH3 are more similar to each other than they are to AMSH2, sharing 47% overall amino acid identity [16,17]. They possess an N-terminal microtubule interacting and transport (MIT) domain and a C-terminal MPN+ domain [15]. Both AMSH1 and AMSH3 interact with endocytosis machinery through the MIT domain [16,17]. Their characteristic MPN+ domain has two zinc ions. One of the two zinc ions activates water molecules to promote the cleavage of isopeptide bonds on Lys 63-linked polyubiquitin chains [23]. Thus, AMSH1 and AMSH3 share similar biochemical properties. AMSH3 was reported to be required for the ETI responses mediated by coiled-coil-type nucleotide-binding leucine-rich repeat receptors (CC-NLRs, CNLs) RPM1 (RESISTANCE TO P. SYRINGAE PV MACULICOLA 1) and RPS2 (RESISTANT TO P. SYRINGAE 2) [19]. Whether AMSH1 also contributes to CNL-mediated resistance remains unclear.

The requirement of the DUB AMSH1 for SNC2-mediated immunity revealed another layer of regulation of SNC2-mediated immunity. The homeostasis of key immune regulator BDA1 seems to determine the SNC2-mediated defense responses output. Given that the MPN+ domain is highly conserved among AMSH proteins, AMSH1 and AMSH3 may share similar function in regulating the protein abundance of other immune regulators functioning in SNC2-mediated pathway.

## 4. Material and Methods

### 4.1. Plant Materials and Growth Conditions

All *Arabidopsis (A.) thaliana* plants used in this study are in the Col-0 ecotype background. The *sard1-1 snc2-1D* mutant was previously described [10]. The *amsh1-1* T-DNA insertion mutant SALK_108544C1 was obtained from *Arabidopsis* Biological Resource Center (ABRC). The *amsh1-2* deletion allele was generated by CRISPR/Cas9. The *A. thaliana* and *N. benthamiana* plants were grown on soil under long-day conditions (16 h light/8 h dark cycle) unless specified. Plants for pathogen-induced gene expression analysis, SA measurement, and infection assays were grown for two weeks under long-day conditions. The seedlings were then transplanted and grown for a further one or two weeks under short-day conditions (8 h light/16 h dark cycle).

### 4.2. Construction of Plasmids

To generate the CRISPR/Cas9 constructs for knocking out *AMSH1*, the genomic sequence of *AMSH1* was subjected to CRISPR-PLANT (http://omap.org/crispr/CRISPRsearch.html, accessed on 13 October 2022) to identify target sequences. The potential off-target was evaluated with Cas-OFFinder (http://www.rgenome.net/cas-offinder/, accessed on 13 October 2022). Target sequences of *AMSH1* were then cloned into the *pHEE401E* vector following the previously described procedure [24].

For transgene complementation, the genomic DNA fragment of *AMSH1* driven by its native promoter was PCR-amplified from wild-type genomic DNA and cloned into *pBasta-3HA* vector to obtain *pAMSH1::AMSH1*. All primers used are listed in Appendix A.

### 4.3. Mutagenesis and Suppressor Screen

Ethyl methane sulfonate (EMS) mutagenesis was performed following a previously described protocol [25]. In brief, around 5000 *sard1-1 snc2-1D* mutant seeds were mutagenized, sterilized, and planted on ½ Murashige and Skoog (½ MS) medium. The 10-day-old well-established seedlings (M1 generation) were transplanted onto soil and grown to maturity. About 40,000 M2 progeny from 2000 M1 plants were grown on soil and screened for plants with increased plant size compared with the *sard1-1 snc2-1D* mutant.

### 4.4. Mapping-by-Sequencing

123-1 mutants were backcrossed with *sard1-1 snc2-1D*. In the F2 population, 36 plants that showed a similar morphology the original 123-1 mutant were selected and pooled for genomic DNA extraction. The genomic DNA was subsequently sequenced using the HiSeq-PE150 platform. Single nucleotide polymorphisms (SNPs) were then analyzed using a variant discovery pipeline based on Genome Analysis Toolkit (GATK) Best Practices run on the Compute Canada Cedar cluster [26].

### 4.5. Gene Expression Analysis

Plants for quantitative reverse transcript-PCR (qRT-PCR) were grown on plates with ½ Murashige and Skoog (MS) and 1% sucrose for 12 days unless specified. To test *Pst* DC3000 *hrcC*-induced gene expression, leaves of 3-week-old plants grown under short-day conditions were syringe infiltrated with a bacterial suspension with OD_600_ of 0.05 in 10 mM MgCl_2_ and collected at 0 h and 12 h post inoculation. The total RNA was extracted using the Plant RNA Mini-Preps Kit (Bio Basic, Markham, ON, Canada). A total of 1 μg of RNA was reverse transcribed into cDNA using the OneScript Reverse Transcriptase (abm, Richmond, BC, Canada). Quantitative real-time PCR was performed using the SYBR Premix Ex Taq II (Takara, Kyoto, Japan). Primers used for amplification of *PR1*, *PR2*, *SARD1*, and *ACTIN1* were described previously [10,27].

### 4.6. Infection Assay

The oomycete infection assays using *Hpa* Noco2 were carried out by spraying the spore suspension in water (50,000 spores/mL) onto 2-week-old seedlings. After 7 days of growth at 18 °C, 4–5 whole above-ground seedling tissue was harvested as a single sample. The tissue was vortexed with sterile water, and spores were counted under a microscope using a hemocytometer.

For bacterial infection assays, two fully extended leaves of the 4-week-old plant grown on soil under short-day conditions were infiltrated with *Pst* DC3000 *hrcC* or *Pst* DC3000 suspension at a dose of OD_600_ = 0.002 or OD600 = 0.0002 in 10 mM MgCl_2_, respectively. Bacterial growth was scored on the day of infection (Day 0) and 3 days post inoculation (Day 3). Two leaf disks from a single plant were collected as one sample. Leaf samples were ground in 10 mM MgCl_2_ solution with glass beads at room temperature. The solution was next diluted and plated on lysogeny broth agar plates with rifampicin antibiotic selection. Bacteria colonies were counted after 36 h incubation at 28 °C and calculated as a colony-forming unit (cfu).

### 4.7. SA Measurement

For pathogen-induced SA accumulation analysis, 4-week-old plants grown under short-day conditions were infiltrated with *Pst* DC3000 *hrcC* at a dose of OD_600_ = 0.05. Samples were harvested at 0 h and 12 h after infiltration. SA was extracted and measured with high-performance liquid chromatography (HPLC) following a previously described procedure [28].

### 4.8. RNAi Construct Design and Transient Expression in N. benthamiana

The RNAi constructs were made using an intron-based method for dsRNA generation [29]. Using the WebApollo LAB3.6 genome browser platform, a gene with two copies, NbL15g10460.1 and NbL03g21980.1, was identified as an *AMSH1*-like gene in *N. benthamiana* (referred to *NbAMSH1* in this study) [30]. The 300 bp target sequence for silencing *NbAMSH1* was selected by the SGN VIGS Tool [31]. The sense and antisense fragments of the target were PCR amplified from wild-type *N. benthamiana* genomic DNA. Intron 3 of *PYRUVATE ORTHOPHOSPHATE DIKINASE* (*PDK*) from *Flaveria trinervia* was used as a spacer sequence between sense and antisense gene fragments to generate a hairpin structure. Then, a sense fragment, the *PDK* intron, and an antisense fragment were accordingly assembled into pCambia1300-35S-E9ter using an NEBuilder^®^ HiFi DNA Assembly Kit (New England Biolabs, Whitby, ON, Canada). The primers used for making RNAi construct were listed in Appendix A.

For transient expression in *N. benthamiana*, empty vector pCambia1300 (OD_600_ = 0.4) and BDA1-HATurboID (OD_600_ = 0.2) were infiltrated into the left halves of four-week-old *N. benthamiana* leaves as control. The RNAi constructs carrying fused sense–intron–antisense fragments of *NbAMSH1* (OD_600_ = 0.4) and BDA1-HATurboID were expressed in the right-half leaves. After infiltration, the plants were kept in dark for 2 days to induce the expression of RNAi constructs. Total protein was extracted from the infiltrated area at 48 h post infiltration and then subjected to immunoblot analysis.

## 5. Conclusions

Overall, our data in this study uncover the role of the DUB AMSH1 in SNC2-mediated immunity. A loss-of-function mutation in *AMSH1* caused partial suppression of *sard1-1 snc2-1D*’s autoimmunity. Characterization of the *amsh1* mutant uncovered a crucial role of AMSH1 in plant basal resistance and PTI responses. The DUB AMSH1 likely regulates PTI through regulating BDA1 protein stability.

## Figures and Tables

**Figure 1 plants-14-00429-f001:**
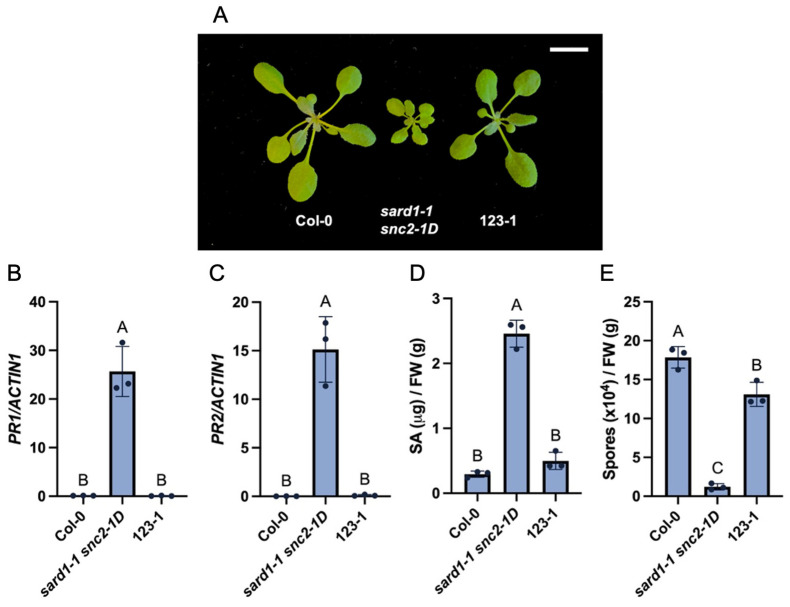
Identification and characterization of the *sard1-1 snc2-1D* suppressor mutant 123-1. (**A**) Morphology of three-week-old soil-grown plants of the indicated genotypes under a long-day condition. Scale bar is 1 cm. (**B**,**C**) Relative expression levels of *PR1* (**B**) and *PR2* (**C**) in the indicated genotypes. Transcript levels were normalized with *ACTIN1*. Error bars represent standard deviations. Letters indicate statistical differences (*p* < 0.05, one-way ANOVA; n = 3). (**D**) Free SA levels in the indicated genotypes. Error bars represent standard deviations. Letters indicate statistical differences (*p* < 0.05, one-way ANOVA; n = 3). (**E**) Growth of *Hpa* Noco2 conidiospores on the indicated genotypes. Error bars represent standard deviations. Letters indicate statistical differences (*p* < 0.05, one-way ANOVA; n = 3).

**Figure 2 plants-14-00429-f002:**
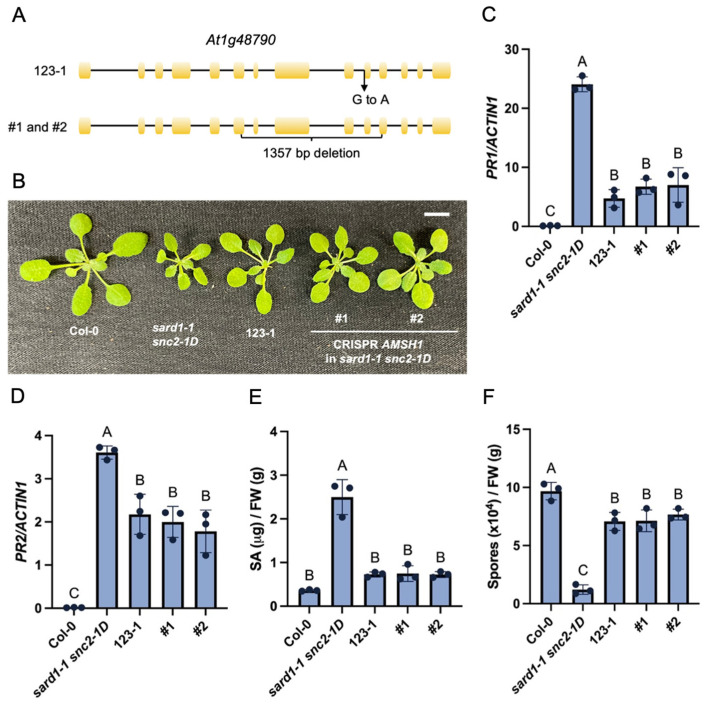
Deletion mutations of *AMSH1* generated through CRISPR suppress the autoimmunity of *sard1-1 snc2-1D.* (**A**) Mutations in *At1g48790*. 123-1 carries a G to A substitution at the splice junction of the tenth exon in *At1g48790* (**top**). #1 and #2 are two independent CRISPR deletion lines of *AMSH1* in the *sard1-1 snc2-1D* background, deleting the same genomic region (**bottom**). (**B**) Morphology of 24-day-old soil-grown plants of the indicated genotypes under long-day condition. Scale bar is 1 cm. (**C**,**D**) Expression levels of *PR1* (**C**) and *PR2* (**D**) in the indicated genotypes as normalized by *ACTIN1*. Error bars represent standard deviations. Letters indicate statistical differences (*p* < 0.05, one-way ANOVA; n = 3). (**E**) Free SA level in the indicated genotypes. Error bars represent standard deviations. Letters indicate statistical differences (*p* < 0.05, one-way ANOVA; n = 3). (**F**) Growth of *Hpa* Noco2 conidiospores on the indicated genotypes. Error bars represent standard deviations. Letters indicate statistical differences (*p* < 0.05, one-way ANOVA; n = 3).

**Figure 3 plants-14-00429-f003:**
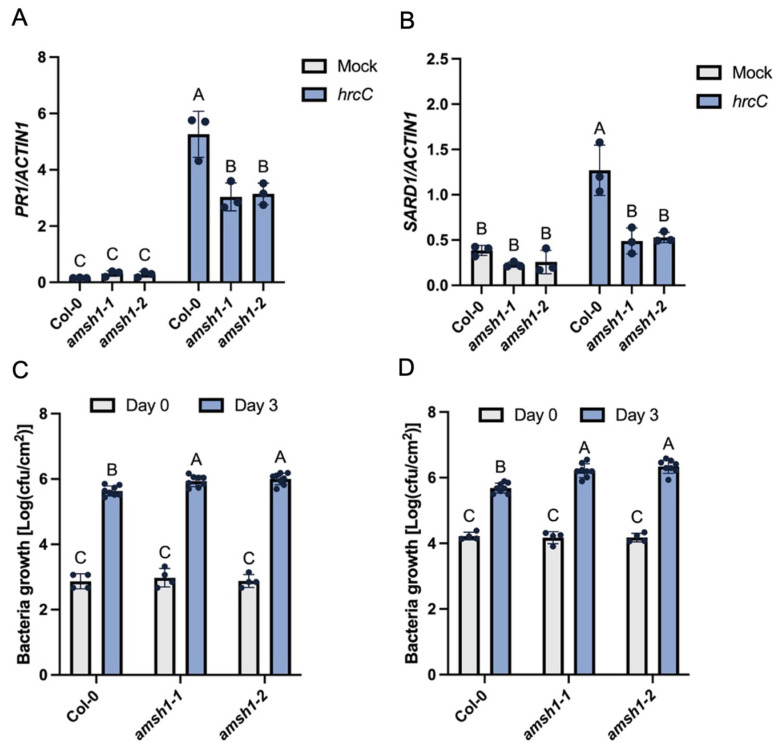
Suppressed PTI and basal immunity in the *amsh1* single mutants. (**A**) Relative expression level of *PR1* in the indicated genotypes treated with 10 mM MgCl_2_ (mock) or *Pst* DC3000 *hrcC* (OD_600_ = 0.05) for 12 h. Values were normalized to the expression of *ACTIN1*. Error bars represent standard deviations. Letters indicate statistical differences (*p* < 0.05, one-way ANOVA; n = 3). (**B**) Relative expression levels of *SARD1* in the indicated genotypes treated with 10 mM MgCl_2_ (mock) or *Pst* DC3000 *hrcC* (OD_600_ = 0.05) for 12 h. Values were normalized to the expression of *ACTIN1*. Error bars represent standard deviations. Letters indicate statistical differences (*p* < 0.05, one-way ANOVA; n = 3). (**C**) Growth of *Pst* DC3000 *hrcC* on the indicated genotypes. Error bars represent standard deviations. Letters indicate statistical differences (*p* < 0.0001, one-way ANOVA; n = 4 for Day 0, n = 8 for Day 3). (**D**) Growth of *Pst* DC3000 on the indicated genotypes. Error bars represent standard deviations. Letters indicate statistical differences (*p* < 0.0001, one-way ANOVA; n = 4 for Day 0, n = 8 for Day 3).

**Figure 4 plants-14-00429-f004:**
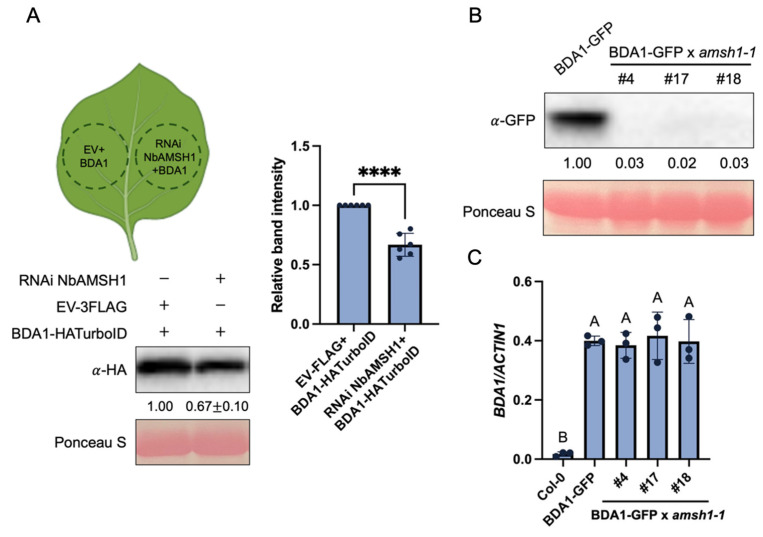
Loss of *AMSH1* leads to decreased protein levels of BDA1 in *N. benthamiana* and *Arabidopsis.* (**A**) Immunoblot analysis of protein levels of BDA1-HA-TurboID in the absence or presence of the RNAi *NbAMSH1* construct in *N. benthamiana*. *Agrobacterium* carrying plasmids expressing BDA1-HA-TurboID together with EV-3FLAG or RNAi *NbAMSH1* were co-infiltrated into one half of each *N. benthamiana* leaf. Samples were collected at 48 h post inoculation (hpi). Equal loading is shown by Ponceau S staining of a non-specific band. The intensity of BDA1-HATurboID bands upon co-infiltration with EV-FLAG was set as 1. Error bars represent standard deviations. **** indicates statistical differences (*p* < 0.05, unpaired t test; n = 6). (**B**) Immunoblot analysis of protein levels of BDA1-GFP in the indicated *Arabidopsis* plants. Equal loading is shown by Ponceau S staining of a non-specific band. (**C**) Relative expression level of *BDA1* in the indicated genotypes. Values were normalized to the expression of *ACTIN1*. Error bars represent standard deviations. Letters indicate statistical differences (*p* < 0.05, one-way ANOVA; n = 3).

**Figure 5 plants-14-00429-f005:**
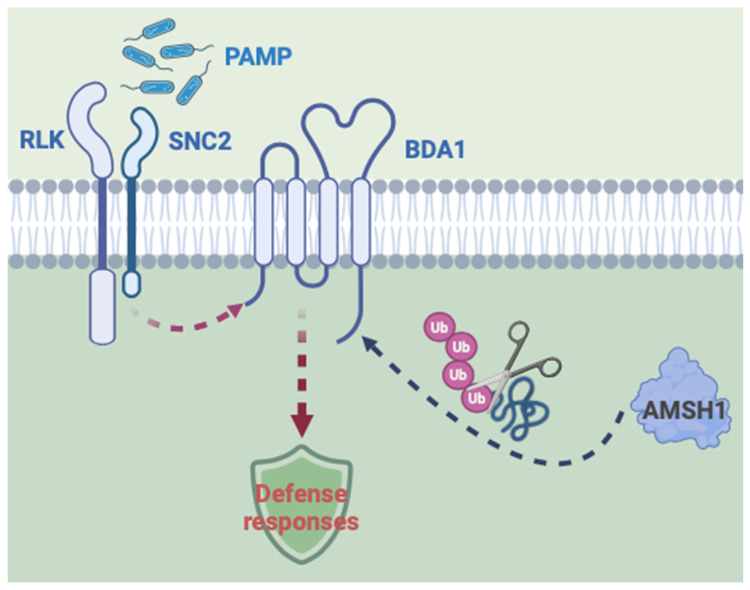
A working model of AMSH1 in SNC2-mediated immunity. The deubiquitinating enzyme AMSH1 contribute to SNC2-mediated immunity by modulating the protein abundance of BDA1.

## Data Availability

All data are presented in the manuscript and Appendix A.

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
