# Peer review of "The Deubiquitinating Enzyme AMSH1 Contributes to Plant Immunity Through Regulating the Stability of BDA1"

_plants, 2025, doi:10.3390/plants14030429_

Round 1

Reviewer 1 Report

Comments and Suggestions for Authors

General Comments

In the manuscript "The Deubiquitinating Enzyme AMSH1 Contributes to Plant Immunity Through Regulating the Stability of BDA1," Yiran Wang and colleagues investigated suppressors of the autoimmune phenotype dependent on the SNC2 mutant. They identified a deubiquitinating enzyme, AMSH1, as a key component in SNC2-mediated immunity. The findings are compelling, suggesting a potential link between AMSH1 and BDA1 as downstream effect of SNC2-mediated immunity.

Major Comments

The reviewer has several concerns regarding the results presented in the manuscript:

1.     In their screen for immunity suppressors in the SNC2 mutant, the authors identified a mutation in AMSH1 resulting in a single amino acid substitution. It is unclear why this polymorphism was not validated as responsible for the autoimmune suppression phenotype.

2.     Additionally, all the results presented are based on a CRISPR-generated deletion mutant of the AMSH1 gene. Do the authors have any hypotheses about the role of this polymorphism in the structure or function of AMSH1?

3.     Furthermore, it has been previously shown that AMSH1 interacts with the ESCRT-III subunit. It would be interesting to know whether the mutation in 123-1 has a similar function.

4.     The discussion section is underdeveloped. It should delve deeper into the findings and explore potential new research pathways.

Author Response

Comment 1. 

  1. In their screen for immunity suppressors in the SNC2 mutant, the authors identified a mutation in AMSH1 resulting in a single amino acid substitution. It is unclear why this polymorphism was not validated as responsible for the autoimmune suppression phenotype.

Mutant 123-1 carries a recessive mutation. Considering most of loss-of-function mutations are recessive, we generated the knock-out allele of AMSH1 in sard1-1 snc2-1D background and found that loss of AMSH1 suppresses the autoimmunity of sard1-1 snc2-1D. Additionally, in Figure S2A and S2B, transforming wild type AMSH1 under its own promoter into the 123-1 mutant restore the autoimmune responses of sard1-1 snc2-1D. These data suggest that the amsh1 mutation in 123-1 is the causal one for the suppressor and AMSH1 is required for the autoimmunity of sard1-1 snc2-1D.

  1. Additionally, all the results presented are based on a CRISPR-generated deletion mutant of the AMSH1 gene. Do the authors have any hypotheses about the role of this polymorphism in the structure or function of AMSH1?

In Figure 3, consistent with the phenotypes of amsh1-2 (a CRISPR deletion mutant of AMSH1), amsh1-1 (a T-DNA insertion allele of AMSH1) also exhibits compromised PTI responses and basal immunity. The G to A mutation of 123-1 mutant affects the splicing of AMSH1 gene and may lead to the nineth intron remaining in the mature mRNA, producing a mutated AMSH1 protein with abnormal C terminal MPN+ domain. Therefore, we hypothesize that the identified polymorphism affects the function of AMSH1 in cleaving the isopeptide bonds on Lys 63-linked polyubiquitin chains. Some details are now added for clarity.

  1. Furthermore, it has been previously shown that AMSH1 interacts with the ESCRT-III subunit. It would be interesting to know whether the mutation in 123-1 has a similar function.

Previous report showed that AMSH1 interacts with the ESCRT-III subunit through its N terminus MIT domain. Since the mutation in 123-1 locates at the splice junction of the tenth exon of AMSH1 and within the C terminus MPN+ domain, we speculate that the mutation in 123-1 probably does not affects the interaction between AMSH1 and ESCRT-III subunit.

 The characteristic MPN+ domain was known to have two zinc ions. One of them activates water molecules to promote the cleavage of isopeptide bonds on Lys 63-linked polyubiquitin chains. The mutation disrupts the RNA splicing of AMSH1 gene and probably affects the cleavage of polyubiquitin chains. These predicted consequences are now added in Discussion.

  1. The discussion section is underdeveloped. It should delve deeper into the findings and explore potential new research pathways.

Thanks for the nice suggestion, discussion on how the polymorphism of AMSH1 affects its function and potential function similarities of AMSH proteins were added.

Reviewer 2 Report

Comments and Suggestions for Authors

The research demonstrates the activity of a deubiquitinase, which is involved in stabilizing transcriptional regulators, which are involved in the plant's immune response.

The approaches taken to resolve the issues at the beginning of the research are appropriate. The following contributions I will make are aimed at improving the manuscript.

Line 46: Define G and R

Lines 47-48: ThePpathogen-Related genes (PR1 and PR2). These are two important genes that were taken into account in the experiments, so I suggest you go into a little more detail about what is known about these genes.

You do not need to capitalize on the factors. You can write them as you do in (ESCRT), line 271, and other parts of the text.

In some parts of the text,  change. ml --> mL

Fig. 1.  Full name of Hpa.

In the same figure in panel E, it is confusing how there are so much more spores in Col-0 than in sard1-1 snC2-1D. In principle there should be more in the mutants due to the lack of impaired immune response or not?

And we find the same case in figure 2 in panel F. How can this behavior be explained?

Finally, I consider it extremely important that you make a diagram showing the location of the deubiquitinase that this work provides. Above all, it can be visualized and quickly in the routes mentioned in the introductory section. It does not have to be very elaborate; I suggest a very simple diagram.

Author Response

The research demonstrates the activity of a deubiquitinase, which is involved in stabilizing transcriptional regulators, which are involved in the plant's immune response.

The approaches taken to resolve the issues at the beginning of the research are appropriate. The following contributions I will make are aimed at improving the manuscript.

Line 46: Define G and R

>> Revised as suggested.

Lines 47-48: The Pathogen-Related genes (PR1 and PR2). These are two important genes that were taken into account in the experiments, so I suggest you go into a little more detail about what is known about these genes. You do not need to capitalize on the factors. You can write them as you do in (ESCRT), line 271, and other parts of the text.

>> Thanks for the suggestion, information about the PR1 and PR2 was added in line 106-108.

In some parts of the text, change. ml --> mL

>> Thanks, revised as suggested.

Fig. 1.  Full name of Hpa.

>> Full name of Hpa was introduced in line 108.

In the same figure in panel E, it is confusing how there are so much more spores in Col-0 than in sard1-1 snc2-1D. In principle there should be more in the mutants due to the lack of impaired immune response or not? And we find the same case in figure 2 in panel F. How can this behavior be explained?

>>A gain-of-function mutation in SNC2 leads to constitutive activation of defense responses in snc2-1D mutant plants and it supports significantly less pathogen growth compared with WT Col-0. SARD1 and CBP60g define two parallel pathways downstream of SNC2, blocking the SARD1 or CBP60g-dependent signaling pathway only partially but not completely affects the constitutive immunes responses in snc2-1D. Therefore, less spores was observed in sard1-1 snc2-1D mutant comparing with the Col-0 plant.

Finally, I consider it extremely important that you make a diagram showing the location of the deubiquitinase that this work provides. Above all, it can be visualized and quickly in the routes mentioned in the introductory section. It does not have to be very elaborate; I suggest a very simple diagram.

>> Thank you for the nice suggestion, a work model of AMSH1 in SNC2-mediated immunity is now added in Figure 5.

Reviewer 3 Report

Comments and Suggestions for Authors

In general the manuscript entittled "The deubiquitinating enzyme AMSH1 contributes to plant immunity through regulating the stability of BDA1" is well written and the experiments well conducted.  Only a minor issue I pbserved

I the results section

2.4. amsh1 Single Mutants Exhibit Compromised PTI Responses and Basal Immunity

The authors described differences in figure 3 that, to me, do not convice me; however, they showed the statistical differences; for example, in panels C and D, the differences between the bars from day 3, the col-0 and amsh1-1/2 values are enough statistically different without dispersion of the replicate values, which is very rare.

from the other results i am fine with the conclutions of the authors

Author Response

In general, the manuscript entitled "The deubiquitinating enzyme AMSH1 contributes to plant immunity through regulating the stability of BDA1" is well written and the experiments well conducted.  Only a minor issue I observed.

In the results section

2.4. amsh1 Single Mutants Exhibit Compromised PTI Responses and Basal Immunity

The authors described differences in figure 3 that, to me, do not convince me; however, they showed the statistical differences; for example, in panels C and D, the differences between the bars from day 3, the col-0 and amsh1-1/2 values are enough statistically different without dispersion of the replicate values, which is very rare.

>> For bacterial infection assays in Figure 3C and 3D, leaf samples were ground in 10 mM MgCl2 solution, and the solution was diluted 3125 times for day 3 group. Finally, the results were presented in Log10(colony-forming unit) scale. The differences between the bars look small in the figure, but the bacterial colonies were around 5-fold difference between the Col-0 and amsh1-1/2 mutants in panel C and around 10-fold difference in panel D due to the log treatment of the counts.

From the other results I am fine with the conclusions of the authors.

Round 2

Reviewer 1 Report

Comments and Suggestions for Authors

Comments

The author's comments and the revised version of the article answer the main referee’s concerns.